# Measurement of the Shear Properties of Extruded Polystyrene Foam by In-Plane Shear and Asymmetric Four-Point Bending Tests

**DOI:** 10.3390/polym12010047

**Published:** 2019-12-30

**Authors:** Hiroshi Yoshihara, Makoto Maruta

**Affiliations:** 1Faculty of Science and Engineering, Shimane University, Nishikawazu-cho 1060, Matsue, Shimane 690-8504, Japan; 2Faculty of Science and Technology, Shizuoka Institute of Science and Technology, Toyosawa 2200-2, Fukuroi, Shizuoka 437-8555, Japan; maruta.makoto@sist.ac.jp

**Keywords:** asymmetric four-point bending (AFPB) test, in-plane shear (IPS) test, extruded polystyrene foam (XPS), shear modulus, shear strength, stress concentration

## Abstract

The shear modulus and shear strength of extruded polystyrene foam were obtained by the in-plane shear and asymmetric four-point bending tests. In addition, the test data were numerically analysed, and the effectiveness of these tests was examined. The numerical and experimental results suggest that the shear modulus and shear strength obtained from the in-plane shear test are significantly smaller than those obtained from the asymmetric four-point bending test because the influence of the stress concentration was less significant. Although the in-plane shear test is standardised in ASTM C273/C273M-11, it is considerable to adopt the asymmetric four-point bending test as another candidate for obtaining the shear properties of extruded polystyrene foam.

## 1. Introduction

At present, extruded polystyrene foam (XPS) is used in the construction of sandwich panels [1,2,3,4], flooring materials, such as tatami mats, used in traditional rooms in Japan [5,6,7,8], and heat-storage tanks [9] and geofoams [10,11,12], because the lightweight nature of XPS is effective to attenuate the seismic forces. To ensure that such construction designs are reliable and cost-effective, it is important to accurately characterize the mechanical properties of XPS, including the shear properties, such as the shear modulus and shear strength.

In a previous study, flexural vibration (FV) tests were conducted to measure the Young’s modulus and shear modulus values of XPS, and its effectiveness was discussed based on numerical and experimental results [13]. In another previous study, torsional vibration (TV) and square-plate twist (SPT) tests were conducted to measure the shear modulus values of XPS [14]. Nevertheless, the tests conducted in these previous studies enable the determination of the shear modulus value alone, and it is impossible to determine the shear strength value. Several methods are considered to measure both the shear modulus and the shear strength values of foam materials. Among them, the in-plane shear (IPS), Arcan, and Iosipescu shear tests are often conducted for several foam materials [15,16,17,18,19,20]. Nevertheless, these methods are not always convenient, because a pair of metallic plates must be bonded or connected to the facings of the foam sample to apply a shearing force. Therefore, the preparation of the specimen is often time-consuming. In particular, there is concern that the stress concentration between the metallic plate and foam sample seriously influences the shear modulus and shear strength values in the IPS test, although the IPS test is standardised in ASTM C273/C273M-11 [19]. Considering these drawbacks, alternate methods should be used to measure the shear modulus and shear strength values of XPS. The asymmetric four-point bending (AFPB) test, which is regarded as an application of the Iosipescu shear test, could help to overcome these drawbacks [21,22,23,24,25,26]. The AFPB test is more advantageous than the aforementioned shear tests in that it does not require an apparatus specially designed for the test but that for the universal four-point bending test; therefore, the test can be conducted easily and conveniently. Despite this advantage, however, there are few examples examining the shear properties of foam materials with the AFPB test [23]. In particular, it is extremely difficult to find any examples conducting the AFPB test using XPS, including the experiment and numerical analyses.

In this study, IPS and AFPB tests were performed on XPS specimens to measure the shear modulus and shear strength values. The validity of the test methods was examined by comparing their results with finite element (FE) calculations. The aim of the study was to use the static test to determine the shear properties of XPS, including the shear modulus and shear strength values.

## 2. Materials and Methods

### 2.1. Specimens

Table 1 shows the XPS panels (STYROFOAM^TM^ series fabricated in The Dow Chemical Company, Tokyo, Japan) used to obtain the test specimens for this study and the nominal densities of the panels [5]. These panels were also used in previous studies [13,14]. The directions along the length, width, and thickness of the XPS panel were defined as the L, T, and Z directions, respectively. The L direction coincided with the extruded direction of the panel. As shown in Figure 1, the initial dimensions of the panels were 910 × 910 × 25 mm in the L, T, and Z directions, respectively. Ten specimens with initial dimensions of 300 × 25 × 25 mm and 170 × 25 × 25 mm were cut from the XPS for the IPS and AFPB tests, respectively. A specimen with its longest dimension coinciding with the L direction was defined as an L-type specimen, whereas one with its longest dimension coinciding with the T direction was defined as a T-type specimen. Therefore, the shear properties in the LT and LZ planes were obtained from the L-type specimen, whereas those in the TL and TZ planes were obtained from the T-type specimen.

The shear modulus values in the LT, LZ, TL, and TZ planes were defined as *G*_LT_, *G*_LZ_, *G*_TL_, and *G*_TZ_, respectively. Additionally, the shear strength values corresponding to these planes were defined as *S*_LT_, *S*_LZ_, *S*_TL_, and *S*_TZ_. To conduct the IPS and AFPB tests, the density of the specimen was measured. After measuring the density, the IPS and AFPB specimens were fabricated by the procedure described below.

### 2.2. In-Plane Shear (IPS) Tests

As described in previous studies [13,14], IPS tests were conducted according to the method based on ASTM C273/C273M-11 [19]. As shown in Figure 2, the specimen was rigidly supported with aluminium plates bonded to the facings using epoxy resin (LOCTITE Easy Mix, cure time = 24 h, Henkel Japan, Yokohama, Japan). The load plate was tapered to a knife-edge and fitted into V-notch loading blocks. A load *P* (N) was applied at the crosshead speed of 1 mm/min until it reached the maximum. The relative displacement between the loading plates *δ* (mm) was measured using a linear variable differential transducer, LVDT (CDP-5M, capacity = 10 mm, Tokyo Sokki Kenkyujo, Tokyo, Japan). In the IPS loading, the shear stress is supposed to be homogeneously distributed. Therefore, the shear stress *τ*_IPS_ was obtained from the following equation:(1)$$\tau_{\mathrm{IPS}}=\frac{P}{TL}$$
and shear strain *γ*_IPS_ were obtained from the following equation:(2)$$\gamma_{\mathrm{IPS}}=\frac{\delta }{B}$$
where *B*, *L*, and *T* are the width, length, and thickness of the specimen, respectively. The shear modulus value was measured from the initial slope of the *τ*_IPS_–*γ*_IPS_ diagram [13,14], whereas the shear strength value was obtained by substituting the maximum load *P*_max_ into Equation (1).

### 2.3. Asymmetric Four-Point Bending (AFPB) Tests

Figure 3a shows the diagram of the AFPB test. As shown in Figure 3b, the shear force is maximum between the inner spans, whereas the bending moment is zero at the mid-span. Therefore, the AFPB test is advantageous to characterize the shear properties in that the shear force is dominant when the failure is induced at the mid-span. This method is regarded as an application of the Iosipescu shear test, which was originally proposed for the measurement of the shear properties of metals [27].

A rectangular bar with the aforementioned dimensions was sandwiched between a pair of moulds with the V-notched configuration: then, it was cut into the shape shown in Figure 3a using a heat wire. To measure the “apparent” shear strain, a biaxial strain gauge (nominal gauge factor = 2.1, gauge length = 1 mm; FCA-1-11, Tokyo Sokki Kenkyujo Co., Ltd., Tokyo, Japan) was bonded at the centre of a side surface. The gauge axes were in directions inclined at ±45° with respect to the *x*-direction. The normal strains in directions inclined at +45° and −45° were defined as *ε*_I_ and *ε*_II_, respectively. In bonding the strain gauge on XPS, there were the following two obstacles:Since the XPS hardly accepted adhesives, the adhesive strength was not often high.Several adhesives often melted the XPS.

Previously to the AFPB tests, several adhesives, including cyanoacrylate, epoxy, and vinyl acetate, were examined to address these obstacles, and, finally, a cyanoacrylate adhesive (CC-35, cure time = 1 h, Kyowa Dengyo, Co., Ltd., Tokyo, Japan) was selected. To promote the adhesive strength, a surface-preparing agent (S-9B, Kyowa Dengyo, Co., Ltd., Tokyo, Japan) was coated on the specimen before using the adhesive. The specimen was eccentrically supported at two trisected points, and the loads were applied at the remaining two points at a crosshead speed of 1.0 mm/min. The distance between the left and right loading points was 150 mm. The shear stress *τ*_AFPB_ is supposed to be homogeneously distributed between the notch roots, and therefore it was obtained from the following equation [22,25,26]:(3)$$\tau_{\mathrm{AFPB}}=\frac{V}{bt}=\frac{P}{2bt}$$
where *b* is the distance between the notch roots, and *t* is the thickness of the specimen. The normal strains in the length and depth directions were defined as *ε_x_* and *ε_y_*, respectively, and the shear strain in the length/depth plane was defined as *γ_xy_*. Then, the normal strains in directions inclined at +45° and −45°, *ε*_I_ and *ε*_II_, respectively, were derived as follows:(4)$$\{\begin{array}{c} \varepsilon_{I}=\frac{\varepsilon_{x}}{2}+\frac{\varepsilon_{y}}{2}+\frac{\gamma_{xy}}{2}\\ \varepsilon_{\mathrm{II}}=\frac{\varepsilon_{x}}{2}+\frac{\varepsilon_{y}}{2}-\frac{\gamma_{xy}}{2} \end{array}$$

Therefore, the “apparent” shear strain measured using strain gauge *γ*_g_ was obtained from the following equations:(5)$$\gamma_{g}=\gamma_{xy}=\varepsilon_{I}-\varepsilon_{\mathrm{II}}$$

The method for obtaining the shear modulus value is described below. In contrast, the shear strength value was obtained by substituting the maximum load into Equation (3).

### 2.4. Tension and Compression Tests for the Strain Gauge Calibration

In the IPS test, the shear strain can be easily measured using an LVDT as described above. In contrast, because it is difficult to set the LVDT in the AFPB test to measure the shear strain in the gauge region, which corresponds to the region between the notch roots, an alternative method for measuring shear strain was required. Optical methods, such as digital image correlation (DIC) and virtual fields method (VFM), are promising in measuring the strain induced in a material with low stiffness, such as foam and paper materials [15,16,17,18,20,28]. In the preliminary tests, it was examined whether the shear strain could be accurately measured using a high-speed digital image sensor (Keyence CV-5000SO, Keyence Corporation, Osaka, Japan), which was effective for measuring the elongation induced during a tensile test of the paper material [29]. On the gauge region of an AFPB specimen, a pair of straight lines, inclined at 45° with respect to the length direction of the AFPB specimen surface, were drawn. Then, the elongation between the lines was photographed using a CCD camera at intervals of 0.5 s and analysed using the high-speed digital image sensor. The shear strain was calculated from dividing the elongation by the initial distance between the lines. In this method, however, the rotation of the lines, which was induced from the large deflection of the AFPB specimen, was significant in the field of view. Additionally, it was difficult to conduct the aforementioned DIC and VFM methods because of the lack of the equipment. Therefore, the optical methods should have been abandoned in this study.

Instead of the optical method, the shear strain was measured using a strain gauge bonded to the gauge region, despite being a classical and provisional method, and the relevancy for using the strain gauge was examined. However, when bonding a strain gauge on a material with a small stiffness such as XPS, the sensitivity of the gauge significantly decreases. As a result, the strain obtained from the strain gauge output is often smaller than the actual strain. Several examples have examined the possibility of using the strain gauge for cellular plastics [30,31]. In this study, the tension and compression tests were conducted for the strain calibration, and it was examined whether the strain obtained from the strain gauge could be transformed into the actual strain. The dimensions of the tension test specimen were 170 × 25 × 25 mm, whereas those of the compression test specimen were 100 × 25 × 25 mm. The length direction coincided with the direction 45° inclined with respect to the L direction because the shear strain in the AFPB test was detected as the normal strains in the ±45° directions, as described above. A strain gauge similar to that used in the AFPB test was bonded to the centre of both LT surfaces. Additionally, a displacement gauge (capacity = 50 mm; PI-5-50, Tokyo Sokki Kenkyujo Co., Ltd., Tokyo, Japan) was attached to the same surfaces. Ten specimens were used for each calibration test. In the tension test, a tensile load was applied to the specimen with a grip the length of which was 35 mm. In the compression test, a compressive load was applied to the end surface of the specimen. The crosshead speed was 1 mm/min, and the strain output from the strain and displacement gauges, defined as *ε*_g_ and *ε*_d_, respectively, were obtained by averaging the strains measured at both LT surfaces. Using the calibrated *ε*_d_–*ε*_g_ relationship, the calibrated shear strain *γ*_c_ was obtained from the shear strain measured from the strain gauge *γ*_g_. The detailed method of the transformation is described below.

## 3. Finite Element Analysis

Two-dimensional finite element analyses (2D-FEAs) were independently conducted on the actual IPS and AFPB tests. The ANSYS 18.2 program was used for the FE analyses. Table 2 presents the elastic properties used in this study. The values of Young’s modulus and shear modulus are similar to those measured in previous studies [13,14]. In these studies, however, the values of the Poisson’s ratios were not measured and but derived as 0.35. Since the values of the Poisson’s ratio were measured from the tension tests in another study [32], they are also used in this study. The model consisted of four-node plane elements. The horizontal and vertical axes of the model were defined as the *x* and *y* directions, respectively.

As shown in Figure 4, the IPS test model consisted of the XPS and aluminium portions. The thickness of the model, *T*, was 25 mm. The horizontal and vertical lengths of the XPS portion, *B* and *L*, respectively, were 25 and 300 mm, respectively. The mesh of the XPS portion was homogeneously divided with dimensions of 2.5 and 5 mm in the *x* and *y* directions, respectively. The number of elements was 1950. Prior to the FEAs using this finite element mesh, other FEAs had been examined using a coarser mesh. Nevertheless, the obtained results were similar to each other, and therefore the finite element mesh shown in Figure 3 was confirmed to be fine enough. In contrast, the aluminium portion had dimensions of 5 and 30 mm in the horizontal and vertical directions, respectively. The elastic moduli of the aluminium were derived as listed in Table 1. The displacement of the bottom edges was restricted in the *x* and *y* directions, whereas the displacement in the *y* direction, defined as *u_y_*, was applied downward to the nodes at the top of the aluminium portion as *u_y_* = 1.0 mm, as shown in Figure 4. Under this boundary condition, the stress components corresponding to each node, defined as *σ_x_*, *σ_y_*, and *τ_xy_*, were obtained. Additionally, the nominal shear stress *τ*_IPS_ and shear strain *γ*_IPS_ were calculated using Equations (1) and (2), respectively. In the FEA, the applied load *P* was obtained from the sum of the reaction forces at the loading points, whereas the relative displacement between the loading plates *δ* was obtained as the value of *u_y_* (=1 mm). The shear modulus value was derived as *τ*_IPS_/*γ*_IPS_, as determined in a previous study [13].

Figure 5 shows the finite element mesh for the AFPB test simulation. The horizontal length of the model was 170 mm, and the model thickness, *t*, was 25 mm. The depth of the model, *H*, was 25 mm, and the distance between the notch roots, *b*, was 7 mm. The mesh was constructed to be finer at the region between the circular notches, as shown in Figure 5b. The number of the elements was 2000. Prior to the FEAs using the IPS simulations, other FEAs had been conducted using a coarser mesh for the AFPB analysis. Nevertheless, the obtained results were similar to each other; therefore, the finite element mesh shown in Figure 5 was also confirmed to be fine enough. The finite element mesh was confirmed to be fine enough as well as that of the IPS model. The nodes corresponding to the locations at *x* = 10 and 110 mm at the bottom of the model (*y* = 0 mm) were restricted, whereas a displacement of 1 mm was applied downward to the nodes corresponding to the locations at *x* = 60 and 160 mm at the top of the model (*y* = 250 mm). The asymmetric loading condition was realised by this boundary condition. Similar to the IPS test simulation, the stress components corresponding to each node, *σ_x_*, *σ_y_*, and *τ_xy_*, were obtained. Additionally, the nominal shear stress *τ*_AFPB_ was calculated using Equation (3), whereas the shear strain *γ*_AFPB_ was obtained from that of the node located at the centre of the model. The shear modulus value was derived as *τ*_AFPB_/*γ*_AFPB_.

## 4. Results and Discussion

### 4.1. Finite Element Analysis

In the IPS test simulation, the stress components corresponding to each node *σ_x_*, *σ_y_*, and *τ_xy_* were normalised by the nominal shear stress *τ*_IPS_ calculated from Equation (1). Figure 6 shows the distribution of the normalised stresses *σ_x_*/*τ*_IPS_, *σ_y_*/*τ*_IPS_ and *τ_xy_/τ*_IPS_ at the mid-width and boundary between the XPS and aluminium plate, which correspond to BB’ and CC’ in Figure 4, respectively. At the mid-width, the shear stress component is more significant than the normal stresses, and its distribution is relatively uniform. However, at the boundary between the XPS and the aluminium plate, the compressive stresses in the *x* and *y* directions are markedly enhanced at point C’ (*σ_x_*/*τ*_IPS_ = −4.47 and *σ_y_*/*τ*_IPS_ = −1.64), because of the rectangular edge between the XPS and aluminium plate. When the shear modulus is measured using an LVDT as determined in the ASTM C273/C273M-11, these stress concentrations and combined stress condition enhance the displacement measured by the LVDT, and the shear modulus value is estimated as low. Additionally, the stress concentrations also enhance the failure. Therefore, there is a concern that the shear strength value is also estimated as low.

In the AFPB test simulation, the *σ_x_*, *σ_y_*, and *τ_xy_* values were normalised by the nominal shear stress *τ*_AFPB_ calculated from Equation (3). Figure 7 shows the distribution of the normalised stresses *σ_x_*/*τ*_AFPB_, *σ_y_*/*τ*_AFPB_ and *τ_xy_/**τ*_AFPB_ at the mid-span and along the bottom notch edge, which correspond to BB’ and CC’ in Figure 5, respectively. The shear stress component is more significant than the normal stresses, but it distributes more uniformly than that in the IPS test. Therefore, when the shear strain is measured at the mid-span, it is expected that the shear modulus value obtained from the actual AFPB test is more precise than that obtained from the actual IPS test. At the notch edge, however, the tensile and compressive stresses in the *x* direction are markedly enhanced as *σ_x_*/*τ*_AFPB_ = −2.40 and 2.36 at the points of *x* = −1.31 and 1.31 mm, respectively. These values are smaller than that at the rectangular edge in the IPS test. However, there is also a concern that the shear strength value is estimated as low because of the combined stress condition. Further research should be conducted on the specimen configuration to measure the shear strength value with reducing the effect of the stress concentration and combined stress condition.

Table 3 shows the shear modulus values obtained from the IPS and AFPB test simulations. Because of the stress concentration described above, the shear modulus values obtained from the IPS test simulations are significantly smaller than those input into the FEM program. In contrast, the shear modulus values obtained from the AFPB simulations are closer to the input values than those obtained from the IPS test simulations. The concentration of the normal stress components is also found in the AFPB test simulation at the notch edge. However, as described above, the stress concentration in the AFPB test is less significant than that in the IPS test. Because the IPS test is standardised as the ASTM C273/C273M-11 [19], it is conducted more frequently than the AFPB test. However, based on the FEA results, it is preferable to measure the shear modulus and shear strength values of XPS from the AFPB test rather than the IPS test.

### 4.2. IPS and AFPB Tests

Figure 8 shows the typical examples of the relationships between the strain measured from the displacement gauge *ε*_d_ and that measured using the strain gauge *ε*_g_ in the tension and compression tests for strain calibration. As shown in this figure, the *ε*_d_–*ε*_g_ relationships obtained from the specimens cut from the same panel coincide well with each other in the tension tests. In contrast, the relationships obtained from the compression tests vary. In the compression test, the load was applied directly to the end surface of the specimen. Therefore, there was a concern that the load was often applied eccentrically to the specimen due to the distortion of the end surface. Because of the variation, it is often difficult to calibrate the strain using the results obtained from the compression tests. In contrast, the load was applied via the grips in the tension tests and the condition of the end surface did not influence the *ε*_d_–*ε*_g_ relationship. Therefore, the *ε*_d_–*ε*_g_ relationship was obtained stably in the tension test. Based on these testing results, the *ε*_d_–*ε*_g_ relationships obtained from the tension tests were used for the calibration in this study.

Because of the insensitivity of the strain gauge, *ε*_g_ is much smaller than *ε*_d_, as shown in Figure 7a. The *ε*_d_–*ε*_g_ relationship is initially linear and gradually becomes concave, and these tendencies were commonly found in every XPS material. Considering these tendencies, the *ε*_d_–*ε*_g_ relationship was formulated using a power function as follows:(6)$$\varepsilon_{d}=\alpha \varepsilon_{g}+{\left(\beta \varepsilon_{g}\right)}^{c}$$
where *α*, *β*, and *c* are the parameters obtained by the method of least squares. The *a*, *b*, and *c* values corresponding to each XPS were obtained from the following procedure:The *ε*_d_–*ε*_g_ relationship corresponding to each specimen was regressed into Equation (6) by the method of least squares.The *ε*_g_ values were virtually determined in the range from 0 to 0.005 at intervals of 0.0001, and the *ε*_d_ values were obtained by substituting the *ε*_g_ values into the regressed equation.The *ε*_d_ values obtained from the same *ε*_g_ value were averaged among the same XPS material. The averaged *ε*_d_–*ε*_g_ relationship was regressed into Equation (6) again, and the *a*, *b*, and *c* values were determined.

The *α*, *β*, and *c* values obtained from this procedure are listed in Table 4. The calibrated shear strain *γ*_c_ was calculated from the shear strain *γ*_g_, which was calculated from the strain gauge output using Equation (6), as follows:(7)$$\gamma_{c}=\alpha \gamma_{g}+{\left(\beta \gamma_{g}\right)}^{c}$$

Figure 9 shows the typical examples of the shear stress–shear strain relationships obtained from the IPS test *τ*_IPS_–*γ*_IPS_, and AFPB test *τ*_AFPB_–*γ*_g_, and *τ*_AFPB_–*γ*_c_. In the AFPB test, the shear strain obtained from the strain gauge *γ*_g_ is much smaller than that in the IPS test *γ*_IPS_ because of the insensitivity of the strain gauge. Additionally, the nonlinearity in the *τ*_AFPB_–*γ*_g_ relationship is not often significant, such that the stress–strain relationship is extremely discrepant from that obtained from the IPS test. When conducting the strain calibration using Equation (6), however, the linear region is significant in the linear portion of the *τ*_AFPB_–*γ*_c_ relationship, and the nonlinear strain region is more pronounced.

Table 5 lists the shear modulus values obtained from the IPS and AFPB tests. Similar to the results of the FEAs, the shear modulus values obtained from the IPS tests are significantly lower than those obtained from the AFPB tests. In the IPS test, the stress concentration at the rectangular edge between the XPS and aluminium plate induces a large shear deformation; therefore, the shear modulus value is measured as lower in the IPS test. In contrast, the stress concentration at the notch edge in the AFPB test is not more significant than that at the rectangular edge between the XPS and aluminium plate in the IPS test.

Table 5 also lists the values of the in-plane shear modulus *G*_LT_ (*G*_TL_) obtained from the static SPT tests in a previous study [14] and the unpaired *t*-tests of the differences of the *G*_LT_ and *G*_TL_ values obtained from these tests were conducted. The SPT tests were conducted using the specimens cut from XPS panels similar to those used in this study. The *G*_LT_ and *G*_TL_ values obtained from the IPS tests are significantly lower than those obtained from the SPT tests in the significance level of 0.01. In contrast, the difference between the *G*_LT_ and *G*_TL_ values obtained from the AFPB and SPT tests are not significant in the significance level of 0.05. Therefore, the AFPB test is promising in obtaining the shear modulus, in a similar way to the SPT test. However, the measurement of the shear strain is indirect, despite the effectiveness in using the strain gauge. To improve the accuracy in measuring the shear modulus, the aforementioned alternative techniques, such as the digital image correlation (DIC) and virtual field method (VFM), should be adopted for the AFPB test in place of the strain gauge [15,16,17,18,20]. Further research should be performed on this topic.

Table 6 lists the shear strength values obtained from the IPS and AFPB tests. The shear strength values obtained from the AFPB tests are significantly higher than those obtained from the IPS tests. Because the effect of the stress concentration is less significant in the AFPB test than in the IPS test, the AFPB test is preferable to the IPS test for measuring the shear strength value.

Figure 10a shows the large deformation at the rectangular edge between the XPS and aluminium plate during the loading. The failure was initiated and propagated from the rectangular edge. The large deformation is induced due to the stress concentration, and, as described above, there is a concern that the shear modulus and shear strength are estimated as low because of the stress concentration at this point. Figure 10b shows the failures induced in the AFPB test. The failure is induced at the top and/or bottom notch edges, and it propagates obliquely in the specimen. It is desirable that the failure is exactly induced at the mid-length, which corresponds to the region between B and B’ in Figure 5b, because the failure at this region is induced due to the pure shear stress, as shown in Figure 7. However, Figure 10b indicates that the catastrophic failure was often induced at a point deviating from the mid-length at the top and/or bottom notch edges. Therefore, there is a concern that failure due to the combined stress condition is induced, as predicted from the FE calculation (Figure 7b). In strongly orthotropic materials like solid wood, even if a failure was initiated at a point deviating from the mid-length, the failure due to the shear stress was also induced at the mid-length when the load was applied continuously [22]. Further research is required on the specimen configuration to reduce the effect of stress concentration thoroughly, although the shear strengths obtained from the AFPB tests are higher than those obtained from the IPS tests as listed in Table 6.

## 5. Conclusions

To measure the shear modulus and shear strength of extruded polystyrene foam (XPS), the in-plane shear (IPS) and asymmetric four-point bending (AFPB) test methods were experimentally and numerically analysed, and the following results were obtained:The results of the FE analyses indicated that the shear modulus values obtained from the AFPB test simulations were closer to the input ones than those obtained from the IPS test simulation. Therefore, it was expected to measure the shear modulus value by the AFPB test accurately.In the actual AFPB tests, the measurement of the shear strain using the strain gauge was indirectly determined from the calibration tests. The in-plane shear modulus values obtained from the AFPB tests were close to those obtained from the square plate twist (SPT) tests conducted in a previous study [14]. Therefore, the AFPB test is effective in measuring the shear modulus. To accurately determine the shear strain, however, alternative methods, such as digital image correlation (DIC) and virtual fields method (VFM), should be adopted in place of the strain gauge.From the FE analyses, the normal stress components were concentrated in a particular region. Therefore, there was a concern that the shear strength value would be estimated as low because the stress concentration enhanced the failure of the specimen. However, the effect of the stress concentration was less in the AFPB test than in the IPS test.Similar to the results obtained from the FE analyses, the experimental results indicated that the shear modulus and shear strength values obtained from the AFPB test were higher than those obtained from the IPS test. Therefore, the effect of the stress concentration was less significant in the AFPB test than in the IPS test.

## Figures and Tables

**Figure 1 polymers-12-00047-f001:**
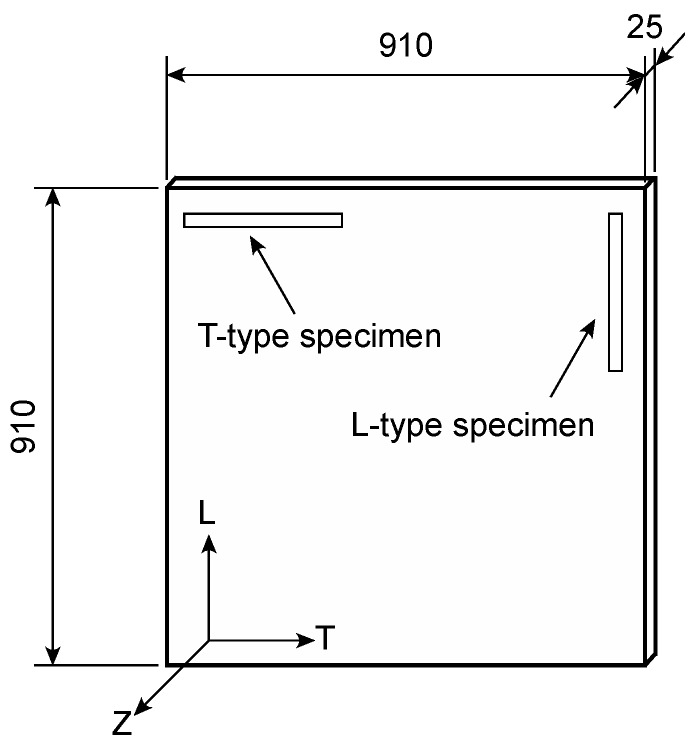
Diagram of the XPS panel. Unit = mm.

**Figure 2 polymers-12-00047-f002:**
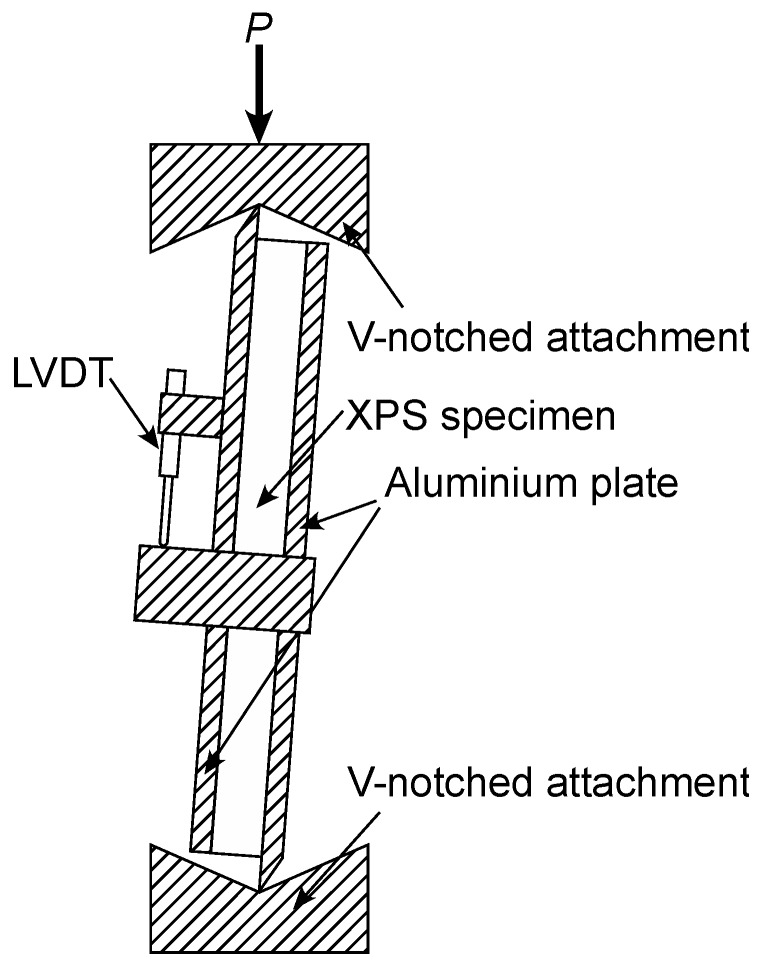
Diagram of the in-plane shear (IPS) test.

**Figure 3 polymers-12-00047-f003:**
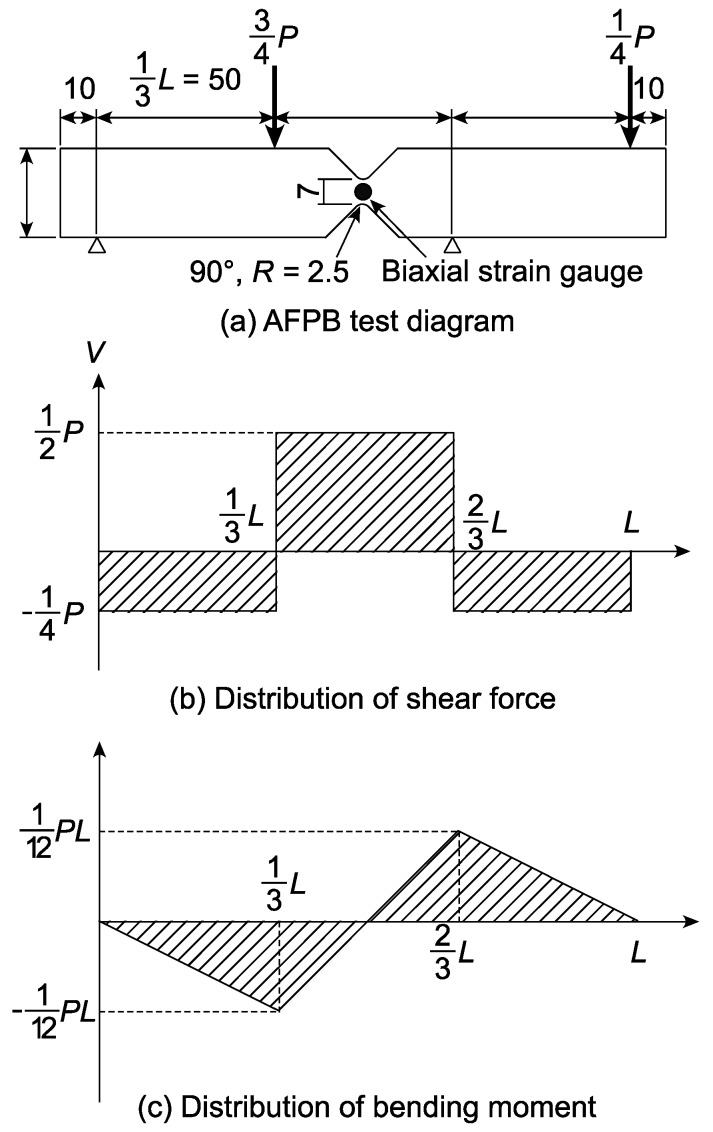
Diagram of the asymmetric four-point bending (AFPB) test.

**Figure 4 polymers-12-00047-f004:**
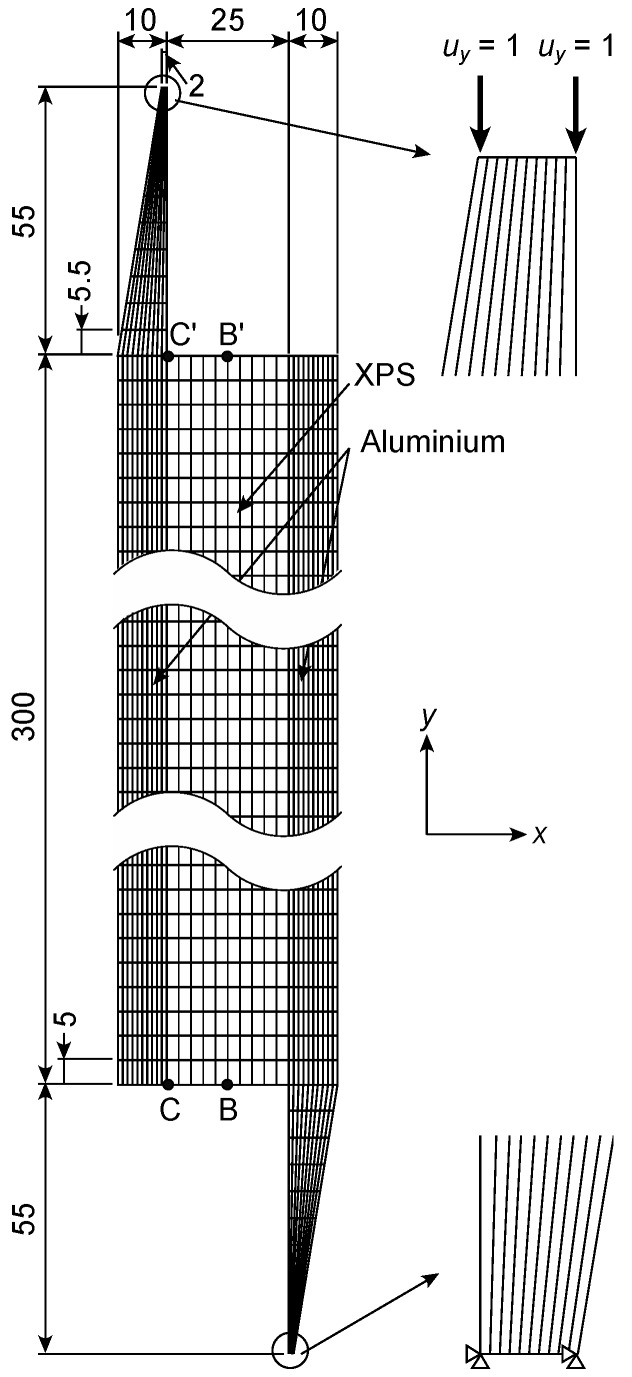
Finite element meshes used in the IPS test simulations and boundary condition. Unit = mm.

**Figure 5 polymers-12-00047-f005:**
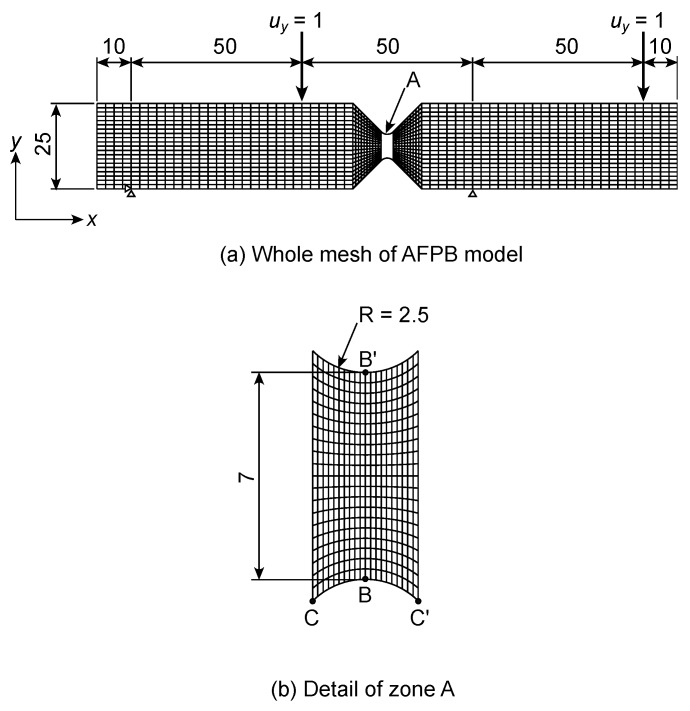
Finite element meshes used in the AFPB test simulations and boundary condition. Unit = mm.

**Figure 6 polymers-12-00047-f006:**
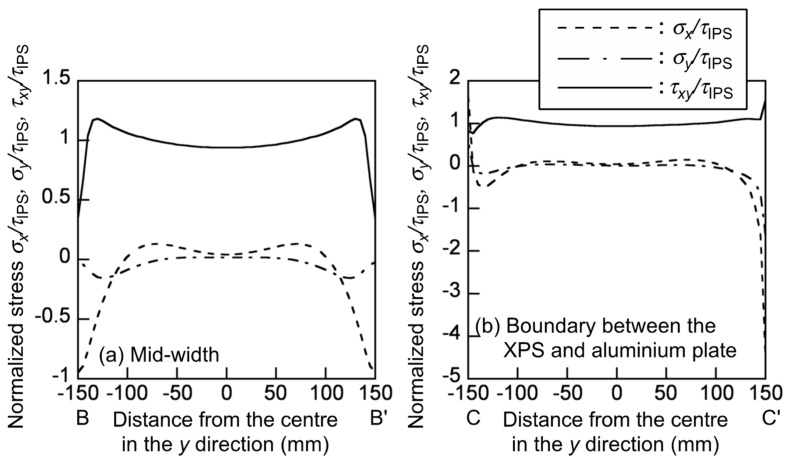
Distribution of the normalised stress *σ_x_/τ*_IPS_, *σ_y_/τ*_IPS_, and *τ_xy_/τ*_IPS_ at (**a**) the mid-width and (**b**) the boundary between the XPS and aluminium plate obtained from the IPS test simulations.

**Figure 7 polymers-12-00047-f007:**
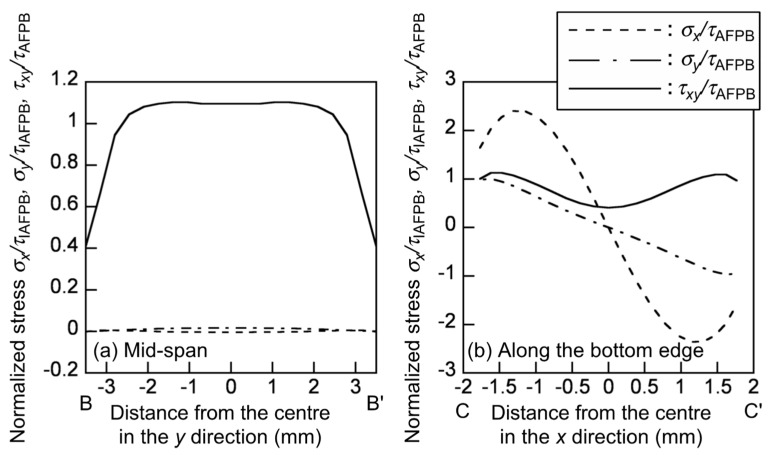
Distribution of the normalised stress *σ_x_/τ*_AFPB_, *σ_y_/τ*
_AFPB_, and *τ_xy_/τ*_AFPB_ at (**a**) the mid-width and (**b**) the boundary between the XPS and aluminium plate obtained from the AFPB test simulations.

**Figure 8 polymers-12-00047-f008:**
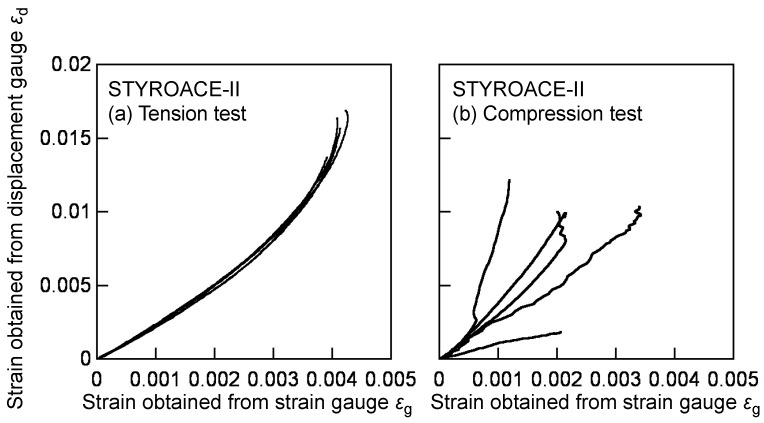
Relationships between the normal strains obtained from the displacement gauge *ε*_d_ and strain gauge *ε*_g_ in the calibration by (**a**) tension and (**b**) compression tests.

**Figure 9 polymers-12-00047-f009:**
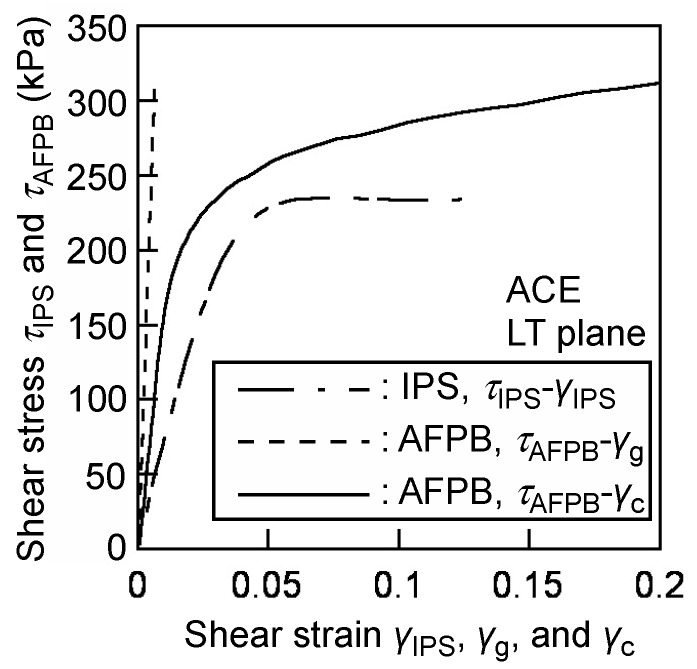
Typical examples of the shear stress–shear strain relationships obtained from the IPS and AFPB test.

**Figure 10 polymers-12-00047-f010:**
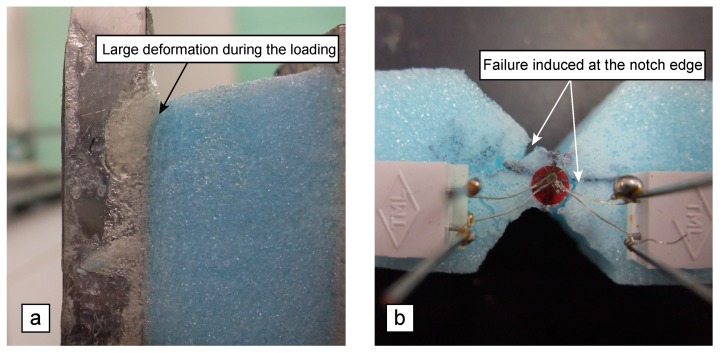
Large deformation of the XPS specimen at the rectangular edge between the XPS and aluminium plate during the loading (**a**), and failures induced at the top and bottom notch edges in the AFPB test (**b**).

**Table 1 polymers-12-00047-t001:** Extruded polystyrene foam (XPS) panels used in this study and their nominal densities.

Material	Code	Density (kg/m^3^)
STYROFOAM IB	IB	26
STYROFOAM B2	B2	29
STYROACE-II	ACE	32
STYROFOAM RB-GK-II	RB-GK	36

**Table 2 polymers-12-00047-t002:** Elastic properties of the XPS and aluminium models used for the FEAs [13,14,32].

Code	Young’s Modulus (MPa)			Shear Modulus (MPa)			Poisson’s Ratio		
	*E* _L_	*E* _T_	*E* _Z_	*G* _LT_	*G* _ZL_	*G* _TZ_	*v* _LT_	*v* _ZL_	*v* _TZ_
IB	17.7	15.7	16.7	6.93	9.42	8.95	0.46	0.53	0.40
B2	24.4	15.9	20.2	7.26	10.3	8.55	0.58	0.44	0.46
ACE	29.0	19.8	24.4	9.18	12.8	10.5	0.53	0.40	0.37
RB-GK	37.5	23.5	30.5	12.1	15.5	13.5	0.43	0.36	0.51
	**Young’s Modulus (GPa)**			**Shear Modulus (GPa)**			**Poisson’s Ratio**		
Aluminium	69.0			27.0			0.28		

**Table 3 polymers-12-00047-t003:** Shear moduli obtained from the IPS and AFPB test simulations by FEM.

Code	*G*_LT_ (MPa)		*G*_LZ_ (MPa)		*G*_TL_ (MPa)		*G*_TZ_ (MPa)	
	IPS	AFPB	IPS	AFPB	IPS	AFPB	IPS	AFPB
IB	5.54	6.16	7.08	8.30	5.58	6.05	6.80	7.77
B2	5.75	6.62	7.65	10.6	5.87	6.21	6.63	7.60
ACE	7.00	9.18	9.08	13.2	7.13	7.87	7.84	8.95
RB-GK	8.69	11.1	10.5	15.5	8.91	10.3	10.4	11.5

Input data of shear modulus: See Table 2.

**Table 4 polymers-12-00047-t004:** *α*, *β*, and *c* values obtained from the regression of *ε*_d_–*ε*_g_ relationship of the tension test data into Equation (6).

Code	*α*	*β*	*c*
IB	3.97	135	5.25
B2	5.20	216	12.6
ACE	2.83	131	8.60
RB-GK	2.18	130	8.95

**Table 5 polymers-12-00047-t005:** Shear moduli obtained from the IPS and AFPB tests.

Code	*G*_LT_ (MPa)		*G*_LZ_ (MPa)		*G*_TL_ (MPa)		*G*_TZ_ (MPa)		*G*_LT_, *G*_TL_ (MPa)
	IPS	AFPB	IPS	AFPB	IPS	AFPB	IPS	AFPB	SPT
IB	4.47	8.00	7.04	9.60	4.64	7.36	6.06	7.47	6.32
	(0.16)	(2.70)	(0.47)	(1.62)	(0.24)	(1.68)	(0.63)	(1.72)	(0.22)
B2	4.96	7.34	7.53	6.76	5.25	6.56	6.43	6.42	6.99
	(0.60)	(1.05)	(0.63)	(0.68)	(0.71)	(0.59)	(0.37)	(0.57)	(0.09)
ACE	6.55	10.4	11.1	16.3	6.93	11.3	8.03	10.6	9.78
	(0.60)	(2.3)	(0.2)	(1.3)	(0.33)	(2.5)	(0.44)	(1.4)	(0.36)
RB-GK	10.3	15.0	12.7	16.6	10.4	11.9	8.96	13.7	13.6
	(0.8)	(2.8)	(0.6)	(3.1)	(0.6)	(2.5)	(1.05)	(3.0)	(0.5)

Results are given as the averages ± (SD). The results of the IPS and static SPT tests are referred from [13,14], respectively.

**Table 6 polymers-12-00047-t006:** Shear strengths obtained from the IPS and AFPB tests.

Code	*S*_LT_ (kPa)		*S*_LZ_ (kPa)		*S*_TL_ (kPa)		*S*_TZ_ (kPa)	
	IPS	AFPB	IPS	AFPB	IPS	AFPB	IPS	AFPB
IB	157	225	208	261	157	213	185	242
	(7)	(2)	(13)	(13)	(12)	(7)	(33)	(14)
B2	150	267	235	298	163	245	196	250
	(14)	(9)	(16)	(30)	(7)	(12)	(16)	(19)
ACE	212	363	289	344	228	271	231	260
	(19)	(45)	(9)	(9)	(11)	(12)	(5)	(27)
RB-GK	282	403	351	429	283	355	278	320
	(36)	(18)	(18)	(40)	(33)	(16)	(31)	(26)

Results are given as the averages ± (SD).

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
