# Peer review of "Measurement of the Shear Properties of Extruded Polystyrene Foam by In-Plane Shear and Asymmetric Four-Point Bending Tests"

_polymers, 2019, doi:10.3390/polym12010047_

Round 1

Reviewer 1 Report

Preliminarily, several adhesives were examined for to address these obstacles, and a cyanoacrylate adhesive (CC-35, cure time = 1 hours, Kyowa Dengyo, Co., Ltd., Tokyo, Japan) was selected.

Why this particular adhesive was selected? Any previous tests, references?

Author Response

To Reviewer 1

Thank you very much for your suggestion. We revised the manuscript and provided a point-by-point to your comment. Followings are the response to your comment:

Preliminarily, several adhesives were examined for to address these obstacles, and a cyanoacrylate adhesive (CC-35, cure time = 1 hours, Kyowa Dengyo, Co., Ltd., Tokyo, Japan) was selected.

Why this particular adhesive was selected? Any previous tests, references?

Answer:

As described in the manuscript, the adhesive was selected from the examination conducted previously to the AFPB test.

Considering your comment, the corresponding sentence (Lines 168-170) is revised to enhance the clarity of description as follows:

Previously to the AFPB tests, several adhesives, including cyanoacrylate, epoxy, and vinyl acetate, were examined for to address these obstacles, and finally, a cyanoacrylate adhesive (CC-35, cure time = 1 hours, Kyowa Dengyo, Co., Ltd., Tokyo, Japan) was selected.

Thank you very much again for your fruitful suggestion. We hope that the revised manuscript is now acceptable for publication. When you think that there are still any insufficient descriptions, however, please inform the issue. We’d like to consider the re-revision as soon as possible.

Yours sincerely,

Hiroshi YOSHIHARA

Faculty of Science and Engineering

Shimane University

Matsue, Shimane

Shimane 690-8504, Japan

Phone: +81-852-32-6508

FAX: +81-852-32-6123

Reviewer 2 Report

Dear Authors,

in my opinion the overall quality of this manuscript is high enough to justify its publication as a technical note in Polymers.

Here below some comments and suggestions:

1) Please remove the website reference at the bottom of Table 1 (The website is not in English Language)

2) Please merge Figs. 10 and 11 in one single Figure.

Best regards

Author Response

To Reviewer 1

Thank you very much for your suggestions and advices. We revised the manuscript and provided a point-by-point to your comments. Followings are the response to your comments:

1) Please remove the website reference at the bottom of Table 1 (The website is not in English Language)

Answer:

Considering your suggestion, the reference is removed in the revised version.

2) Please merge Figs. 10 and 11 in one single Figure.

Answer:

Considering your advice, Figs. 10 and 11 are merged and denoted as Figs 10(a) and (b), respectively, in the revised version. The corresponding descriptions in the manuscript are also revised.

Thank you very much again for your fruitful suggestions and advices. We hope that the revised manuscript is now acceptable for publication. When you think that there are still any insufficient descriptions, however, please inform the issue. We’d like to consider the re-revision as soon as possible.

Yours sincerely,

Hiroshi YOSHIHARA

Faculty of Science and Engineering

Shimane University

Matsue, Shimane

Shimane 690-8504, Japan

Phone: +81-852-32-6508

FAX: +81-852-32-6123
